

**1**     **Predominance of hexamethylated 6-methyl branched glycerol dialkyl glycerol**

**2**     **tetraethers in the Mariana Trench: Source and environmental implication**

**3**     Wenjie Xiao[1], Yasong Wang[1], Yongsheng Liu[1], Xi Zhang[1], Linlin Shi[1], Yunping Xu[1*]

**4**     [1] Shanghai Engineering Research Center of Hadal Science & Technology, College of Marine Sciences,

**5**     Shanghai Ocean University, Shanghai 201306, China

**6**     *Correspondence. Yunping Xu (ypxu@shou.edu.cn)

**8**     **Abstract.** Branched glycerol dialkyl glycerol tetraethers (brGDGTs) are useful molecular indicators

**9**     for organic carbon (OC) source and paleoenvironment. Their application in marine environments,

**10**     however, is complicated because of the mixed terrestrial and marine contributions to brGDGTs.

**11**     Here, we employ two dimensional (2D) ultrahigh-performance liquid chromatography-mass

**12**     spectrometry (UHPLC-MS) to analyze brGDGTs in sediments from the Challenger Deep, Mariana

**13**     Trench, the deepest ocean in the absent of terrestrial influence. The unique feature is the absence of

**14**     5-methyl brGDGTs, and the strong predominance of hexamethylated 6-methyl brGDGT (IIIa')

**15**     (73.4±2.4% of total brGDGTs). The brGDGTs-reconstructed pH is 8.22±0.07, close to seawater pH.

**16**     This, combined with characteristics of $\delta^{13}C$ (–19.82±0.25%), OC/TN ratio (6.72±0.84), branched

**17**     and isoprenoid tetraether (BIT) index (0.03±0.01) and the acyclic hexa-/pentamethylated brGDGTs

**18**     ratio (7.13±0.98), strongly suggest that brGDGTs are of autochthonous products from benthic

**19**     bacteria or planktonic bacteria. The compiling of literature data reveals that enhanced fractional

**20**     abundance of hexamethylated 6-methyl brGDGTs is common in diverse continental margins when

**21**     the marine influence became intensified. This may reflect an adaption of brGDGTs-producing

**22**     bacteria to weak alkaline seawater and low ambient temperature. Based on the global dataset, the

**23**     cross plot of acyclic hexa-/pentamethylated brGDGTs ratio and fractional abundance of brGDGT-

**24**     IIIa' is an effective approach to distinguish the terrestrial vs. marine provenance of brGDGTs.

**26**     **1.    Introduction**

**27**         Glycerol dialkyl glycerol tetraethers (GDGTs) are widely distributed biomarkers in terrestrial

**28**     and marine settings (Schouten et al., 2013 and references therein). There are two major types of

**29**     GDGTs, isoprenoidal GDGTs (iGDGTs) and branched GDGTs (brGDGTs) (Sinninghe Damsté et



al., 2000; Weijers et al., 2006). IGDGTs containing isoprenoid carbon skeleton are biosynthesized
by archaea such as Thaumarchaeota, Crenarchaeota and Euryarchaeota (Sinninghe Damsté et al.,
2002; Schouten et al., 2008; Knappy et al., 2011; Lincoln et al., 2014). In contrast, brGDGTs
consisting of 4–6 methyl groups and 0–2 cyclopentane moieties are biosynthesized by certain
bacteria including, but not limit to, Acidobacteria (Sinninghe Damsté et al., 2011). These bacteria
are able to alter the degree of methylation and cyclization of brGDGTs with changing ambient
environmental conditions (Weijers et al., 2007b). A survey for global soils reveals that the
Cyclization of Branched Tetraethers (CBT) correlates with soil pH, while the Methylation of
Branched Tetraethers (MBT) is dependent on mean annual air temperature (MAT) and to less extent
on soil pH (Weijers et al., 2007b; De Jonge et al., 2014a), leading to the development of brGDGTs-
based MBT/CBT proxies for paleo-pH and MAT. The concentration of brGDGTs is substantially
higher in peats and soils than marine sediments, and generally decreases from coastal to distal
marine sediments (Hopmans et al., 2004; Schouten et al., 2013). These distribution patterns support
that brGDGTs in marine settings is derived from terrestrial (particularly soil) inputs. Consequently,
the Branched vs. Isoprenoid Tetraether (BIT) index was proposed for estimation of terrestrial (soil)
OC in marine sediments (Hopmans et al., 2004).

For the past two decades, the brGDGT-derived proxies such as BIT, MBT and CBT have been

increasingly used to assess OC source (Herfort et al., 2006; Kim et al., 2006; Loomis et al., 2011;
Wu et al., 2013), soil pH and MAT in a diverse of environments (Weijers et al., 2007a; Sinninghe
Damsté et al., 2008; Peterse et al., 2012; Yang et al., 2014). However, the weakness of brGDGTs-
based proxies is their source uncertainty. Although brGDGTs were assumed to be specific for
soil/peat bacteria, distinct compositions of brGDGT in rivers (Zhang et al., 2012; Zell et al., 2013;
Zell et al., 2014a), lakes (Sinninghe Damsté et al., 2009; Tierney and Russell, 2009; Loomis et al.,
2011; Buckles et al., 2014), marine waters (Liu et al., 2014; Xie et al., 2014; Zell et al., 2014b) and
sediments (Peterse et al., 2009; Zhu et al., 2011; Xiao et al., 2016) support multiple sources of
brGDGTs.

The employment of one liquid chromatography (LC) column identified nine individual

brGDGTs, all of which were assigned as 5-methyl brGDGTs (Schouten et al., 2007). By improving
the performance of liquid chromatographic separation, De Jonge et al. (2013) found that the peaks



previously identified as 5-methyl brGDGTs were actually the coeluted mixtures of 5-methyl and 6-
methyl brGDGTs (3 hexa- and 3 pentamethylated 6-methyl brGDGTs). As a result, the number of
identified brGDGTs increases from 9 to 15, which are further expanded after identification of 7-
methyl brGDGTs and other isomers (Ding et al., 2016). The analytical improvement has opened the
window for the redefinition and recalibration of brGDGT-based proxies and reassessment of
brGDGT sources (De Jonge et al., 2014a; Xiao et al., 2015). Adopting the new chromatographic
method, several studies provide the clues of in-situ production of brGDGTs in rivers (De Jonge et
al., 2014b; De Jonge et al., 2015), lakes (Weber et al., 2015; Weber et al., 2018) and marine
sediments (De Jonge et al., 2016; Sinninghe Damsté, 2016). For example, De Jonge et al. (2014b)
found that the brGDGT distribution in suspended particulate matter (SPM) of the Yenisei River is
fairly constant and characterized by high abundance of brGDGT-IIIa', which were different from
that in surrounding soils. An extended study also by De Jonge et al. (2015) showed a marked shift
of brGDGTs' compositions from SPM of the Yenisei River to sediments of the Kara Sea. Sinninghe
Damsté (2016) reported brGDGTs in surface sediments from the Berau River delta (Kalimantan,
Indonesia), and suggested in-situ brGDGT production in coastal settings based on the number of
cyclopentane rings ($\#ring_{tetra}$). It should be pointed out that all these studies paid attention to rivers
and continental margins (e.g., De Jonge et al., 2015; Sinninghe Damsté, 2016; Warden et al., 2016),
where the multiple sources and complex processes make difficulty in discerning allochthonous
terrestrial vs. autochthonous marine contributions to the brGDGT pool. Therefore, open ocean in
absence of terrestrial influence is an ideal venue for assessment of source and characters of
brGDGTs in marine settings.
Here, we choose the Challenger Deep, Mariana Trench to analyze brGDGTs in marine
sediments. This deepest trench (ca. 11000 m) is remote from any mainland, and has no significant
terrestrial influence (Jamieson, 2015). Our goals are two folds: 1) to determine the composition and
concentration of brGDGTs in the Mariana Trench sediments and constrain their source; and 2) to
characterize in-situ produced brGDGTs in marine sediments and assess their environmental
implication at the global scale by compiling literature data.

**2. Material and methods**



## 2.1 Study area and sampling

The Mariana Trench is formed as the subduction of Pacific plate beneath the eastern edge of the Philippine Sea plate. It has a total length of ca. 2500 km and a mean width of 70 km (Fryer, 1996). The deepest point, the Challenger Deep, is located in southern rim of the Mariana Trench and has the water depth of ca. 11000 m. Owing to high current speeds and variable current directions, sediment erosion and/or resuspension at the sediment-water interface may frequently occur (Taira et al., 2004; Turnewitsch et al., 2014). The Mariana Trench is remote from the landmass and located in the extremely oligotrophic Pacific Gyre with annual primary production rate of ca. 59 g C m$^{-2}$ y$^{-1}$ (Jamieson, 2015). Consequently, the sinking fluxes of particulate OC is low. However, the sediment of the Challenger Deep was found exhibiting intensive, microbially-mediated biogeochemical recycling processes relative to that of adjacent abyssal plains (Glud et al., 2013). Such character has been attributed to unique "V"-shaped geometry, intense seismic activity and high-frequency fluid dynamics within the trench that promotes lateral transport of sediments from surrounding shallow regions and accumulation of sedimentary organic matter in trench bottom (Jamieson, 2015; Xu et al., 2018).

During an expedition aboard RV Zhangjian (Dec. 2016 to Feb. 2017), a sediment core (MT1, 11.43 °N, 142.36 °E, water depth 10840 m, core length 11 cm) was retrieved in the Challenger Deep using an autonomous 11000 m-rated lander (Fig. 1). The core was immediately stored at –20 °C in a dark room on board until transported to the laboratory in Shanghai (China) where the core was sliced at 1–2 cm interval and kept in a –25 °C freezer. Prior to analysis, all sliced sediment samples ($n$ = 10) were freeze dried at –40 °C and homogenized by steel spatulas.

## 2.2 Lipid extraction and GDGT analyses

Sediment samples (0.5–2 g) were mixed with known amount of C$_{46}$ GDGTs (internal standard) and 15 ml of mixed dichloromethane/methanol (3:1 v/v). After ultrasonically extracted for 15 min, the extracts were centrifuged (3000 rpm, 5 min) and the supernatants were decanted into clean flasks. The extraction was repeated three times. The combined extracts were concentrated by a Rota Evaporator and further blown down to dryness under mild nitrogen streams. The total lipid extract was dissolved in hexane/isopropanol (99:1, v/v) and filtered through a 0.45 μm PTFE filter prior to



analysis. An Agilent ultrahigh performance liquid chromatography-atmospheric pressure chemical
ionization–triple quadruple mass spectrometry system (UHPLC-APCI-MS) was used. The
separation of 5- and 6-methyl brGDGTs was achieved with two silica LC columns in sequence (150
mm × 2.1 mm; 1.9 μm, Thermo Finnigan; USA). The concentration of individual GDGTs was
determined by comparison of the respective protonated ion peak areas with $C_{46}$ GDGT in a selected
ion monitoring (SIM) mode. The protonated ions were m/z 1050, 1048, 1046, 1036, 1034, 1032,
1022, 1020 and 1018 for brGDGTs, 1302, 1300, 1298, 1296 and 1292 for iGDGTs and 744 for $C_{46}$
GDGT.

**2.3  GDGT-derived parameters**
The BIT index, an abundance ratio of acyclic hexa- to pentamethylated brGDGTs and the
weighted average number of cyclopentane moieties for the tetramethylated brGDGTs ($\#rings_{tetra}$)
were calculated according to the definitions of Hopmans et al. (2004), Xiao et al. (2016) and
Sinninghe Damsté (2016), respectively. The roman numbers denote relative abundance of GDGTs
that are depicted in Fig. 2.
$BIT=(Ia+IIa+IIIa+IIa'+IIIa')/(Ia+IIa+IIIa+IIa'+IIIa'+Cren)$                (1)
$\sum IIIa / \sum IIa = (IIIa+IIIa')/(IIa+IIa')$                (2)
$\#rings_{tetra}=(Ib+2*Ic)/(Ia+Ib+Ic)$                (3)
pH was reconstructed using the CBT' index, while MAT was calculated according to the
definition of a Multiple linear Regression-based MAT ($MAT_{mr}$) (De Jonge et al., 2014a).
$CBT'=\log[(Ic+IIa'+IIb'+IIc'+IIIa'+IIIb'+IIIc')/(Ia+IIa+IIIa)]$                (4)
$MAT_{mr}=7.17+17.1*Ia+25.9*Ib+34.4*Ic − 28.6*IIa$                (5)

**2.4  Bulk geochemical analysis**
About 1–2 g of each sediment sample was treated with 1 N HCl for three days at room
temperature to remove carbonates, rinsed into neutral pH and freeze-dried. After homogenized with
an agate mortar and pestle, approximately 35–40 mg of decarbonated sediments were weighed and
analyzed using a model 100 isotope ratio mass spectrometer (IsoPrime Corporation, Cheadle, UK)
and a Vario ISOTOPE cube elemental analyzer (Elementar Analysensystem GmbH, Hanau,





Germany). All isotopic data were reported in δ notation relative to VPDB. The intra-lab standards
for normalizing stable carbon isotopic composition ($\delta^{13}$C) was USG24 (Graphite, –16.05‰), which
was obtained from the International Atomic Energy Agency (IAEA, Vienna, Austria). The average
standard deviation of each measurement, determined by replicate analyses of two samples, was
±0.004 wt% for organic carbon (OC) content, ±0.031 wt% for total nitrogen (TN) content and ±0.03‰
for $\delta^{13}$C.

**2.5 Literature data compilation**
The dataset in this study is composed of relative abundance of brGDGTs from 2031 samples,
including 634 soil samples, 473 peat samples, 88 river samples, 410 lake samples and 426 marine
samples (Fig. 1). The detailed information about these samples was listed in supplementary material.
The soil samples are from globally distributed soils (De Jonge et al., 2014a; Ding et al., 2015; Xiao
et al., 2015; Yang et al., 2015; Lei et al., 2016; Wang et al., 2016; Li et al., 2018; Wang et al., 2018;
Zang et al., 2018; Wang et al., 2019). The peat samples are from 96 different peatlands around the
world (Naafs et al., 2017). The river samples are from Danube River (Freymond et al., 2016),
Yenisei River (De Jonge et al., 2015) and Tagus River (Warden et al., 2016). The lake samples are
from East African lakes (Russell et al., 2018), Chinese lakes (Dang et al., 2016; Li et al., 2017; Dang
et al., 2018), Lake St Front (Martin et al., 2019), Lake Lugano and other lakes in the European Alps
(Weber et al., 2018). The marine samples are from Atlantic Ocean (Warden et al., 2016), Kara Sea
(De Jonge et al., 2015; De Jonge et al., 2016), Berau River delta (Sinninghe Damsté, 2016), Ceará
Rise (Soelen et al., 2017), North Sea (Dearing Crampton-flood et al., 2018), and Mariana Trench in
this study. The criteria for citing the literature data is that both 5- and 6-methyl brGDGTs should be
separated and quantified. It is noted that two studies (Weber et al., 2018; Martin et al., 2019) have
analyzed 5-, 6- and 7-methyl brGDGTs. But due to very limited reports for 7-methyl brGDGTs,
these compounds are not included in this study.

**2.6 Statistical analysis**
The SPSS package 22 (IBM, USA) was used for statistical analyses including Pearson
correlation coefficient ($r$) and one-way Analysis of Variance (ANOVA). The significance level was



set at $P < 0.05$ unless stated elsewhere.

**3.    Results**
**3.1    Bulk geochemical parameters**
The OC content, TN content, molar ratio of OC and TN content (OC/TN) and $\delta^{13}C$ value of
sediments from the Challenger Deep are summarized in Table 1. The OC and TN contents of
sediments vary between 0.26% and 0.31% (0.28±0.01%; mean±STD; same hereafter) and between
0.04% and 0.06% (0.05±0.01%), respectively. The OC/TN and $\delta^{13}C$ values range from 5.62 to 8.34
(6.72±0.84) and –19.47‰ to −20.27‰ (–19.82±0.25%), respectively. Both the $\delta^{13}C$ and OC/TN
values are comparable to previously reported data for the southern Mariana Trench rim and slope
($\delta^{13}C$, –20.48±0.88%; OC/TN, 7.00±1.76) (Luo et al., 2017).

**3.2    Concentration and composition of GDGTs in the Mariana Trench**
The concentration of iGDGTs and brGDGTs are summarized in Table 2. The summed
concentration of total GDGTs in sediments of the Mariana Trench varies from 574 to 1162 ng g$^{-1}$
dry weight sediment (dws) (873±166 ng g$^{-1}$ dws), corresponding to 308±56 μg g$^{-1}$ OC. The
crenarchaeol is the dominant GDGTs at the concentration of 353 to 667 ng g$^{-1}$ dws (533±99 ng g$^{-1}$
dws), corresponding to 188±33 μg g$^{-1}$ OC. The concentration of brGDGTs ranges from 11 to 18 ng
g$^{-1}$ dws (15±3 ng g$^{-1}$ dws), corresponding to 5±1 μg g$^{-1}$ OC and much lower than the concentration
of iGDGTs. As a result, the BIT index is low in all samples with an average value of 0.03±0.01.
Our improved chromatography has achieved a full separation of 5- and 6-methyl brGDGTs.
Interestingly, only a single peak was detected on the mass chromatogram of acyclic penta- (m/z
1036) and hexamethylated (m/z 1050) brGDGTs (Fig. 3). This feature is distinct difference from
previous studies that have identified two or more peaks (5-methyl, 6-methyl and even 7-methyl
isomers) (e.g., De Jonge et al., 2013; Xiao et al., 2015; Ding et al., 2016). In order to determine the
structure of brGDGTs in the Mariana Trench sediments, we take advantage of an acidic soil sample
from China (Soil-1). This sample was identified to contain both 5-methyl brGDGTs (major) and 6-
methyl brGDGTs (minor) (Xiao et al., 2015), and have the IIIa/IIIa' and IIa/IIa' ratios of 12.5 and
8.2, respectively (Fig. 3a, b). After combining Soil-1 (soil) and MT-4 (Mariana Trench), two peaks





were detected for m/z 1050 (hexamethylated brGDGTs) as well as m/z 1036 (pentamethylated
brGDGTs) (Fig. 3e, f). The comparison of retention time among Soil-1, MT-4 and the combined
sample of Soil-1 and MT-4 shows that the peaks of m/z 1050 and 1036 in the MT-1 are
pentamethylated 6-methyl brGDGTs (IIa') and hexamethylated 6-methyl brGDGTs (IIIa'),
respectively, eluting after 5-methyl brGDGTs from Soil-1 (Fig. 3). This assignment was
corroborated by the reduced 5-emthyl/6-methyl brGDGT ratio of the combined sample that is 1.4
for m/z 1050 and 7.4 for m/z 1036 (Fig. 3e, f).
Throughout the sediment core, the brGDGTs are constantly dominated by 6-methyl isomers
(82.25–86.91%). The fractional abundance of 5-methyl brGDGTs, however, was too low to be
quantified. For individual compounds, brGDGT-IIIa' is the most abundant (73.40±2.39% of total
brGDGTs), followed by brGDGT-Ia (12.46±1.14%) and brGDGT-IIa' (10.45±1.20%). The cyclic
compounds (brGDGT-Ib, Ic, IIb') are minor constituents of the brGDGTs (3.69±0.75%), resulting
in low #rings$_{tetra}$ values (0.26±0.04). The classification based on the number of methyl groups shows
the dominance of hexamethylated brGDGTs (73.53±2.56%) over tetramethylated (15.43±1.53%)
and pentamethylated (11.04±1.49%) brGDGTs.

**4. Discussion**
**4.1 In-situ production of 6-methyl brGDGT in the Mariana Trench**
To the best of our knowledge, there are only two reports about GDGTs in the Mariana
subduction zone. Guan et al. (2019) investigated iGDGT distribution in the surface sediments
(4900–7068 m) from the southern Mariana Trench, while Ta et al. (2019) analyzed iGDGTs and
brGDGTs in two sediment cores (ca. 5400 m) at subduction plate of the Mariana Trench. These two
studies, however, did not separate the 5- and 6-methyl brGDGTs, and thus are unable to reveal any
information about source and environmental implication of 5- and 6-methyl brGDGTs. In our study,
the strong predominance of 6-methyl brGDGTs and the absence of 5-methyl brGDGTs in the Marine
Trench sediments are a unique feature. In order to understand the mechanism to produce such unique
compositions of brGDGTs, source assessment of brGDGTs is needed.
The multiple lines of evidence from stable carbon isotope, OC/TN ratio and biomarkers
unanimously support an in-situ production of brGDGTs in the Mariana Trench. The $\delta^{13}$C and OC/TN



ratio have been widely used to distinguish terrestrial vs. marine OC (Meyers, 1997). Generally,
marine algae and bacteria are protein-rich and have OC/TN ratio of 4 to 10, whereas vascular land
plants are cellulose and lignin-rich and have OC/TN ratio of 20 or greater. Due to different carbon
sources and photosynthetic pathways, the typical $\delta^{13}C$ value is ca. –22‰ to –20‰ for marine
organisms (Meyers, 1994) and –27‰ for terrestrial $C_3$ plants (O'Leary, 1988). Sediments from the
Mariana Trench yield enriched $\delta^{13}C$ signatures (–19.82±0.25‰) and low OC/TN values (6.72±0.84),
suggesting marine phytoplankton/bacteria as a major contributor to sedimentary OC (Fig. 4). This
result is expected since the Mariana Trench is remote from the landmasses (Fig. 1) and also agrees
with the previous report from Luo et al. (2017).

Long-distance dust transport from continent to open ocean might deliver brGDGTs to the

Mariana Trench. Unfortunately, no data is available about brGDGTs of eolian dust in the Mariana
Trench region. Weijers et al. (2014) compared the composition of brGDGTs between the marine
sediments and atmospheric dust in the equatorial West African coast, and the great difference
suggests an in-situ production of brGDGTs in the marine sediments, rather than dust input. Here,
we examine the brGDGT compositions in the Mariana Trench sediments with literature data from
global environmental settings (Fig. 5). Relative to the Mariana Trench sediments (brGDGT-Ia
12.46±1.14%, 5-methyl brGDGTs ~0, brGDGT-IIIa' 73.40±2.39%), terrestrial samples are
characterized by significantly higher proportions of brGDGT-Ia (soil 37.52±25.91%, peat
59.40±21.19%, river 15.38±2.97%) and 5-methyl brGDGTs (soil 23.56±14.83%, peat
34.04±19.18%, river 33.25±8.51%), but lower proportions of brGDGT-IIIa' (soil 4.89±4.82%, peat
4.86±4.68%, river 11.68±4.40%) ($p < 0.005$) (Fig. 5). These terrestrial samples are globally
distributed and many of them are from inner Asian continent, the major source area of dust in North
Pacific (Husar et al., 2001). Thus, brGDGTs in the Mariana Trench sediments are unlikely derived
from air dusts. We note that brGDGTs in the Lake Lugano, a deep meromictic Swiss lake, is also
characterized by the strong predominance of brGDGT-IIIa' (up to 90%) (Fig. 5; Weber et al., 2018),
where the distributional patterns and $\delta^{13}C$ of brGDGTs support an provenance in the lower part of
the oxygenated water column. However, most marine samples in the literature present low
proportions of brGDGT-IIIa' (9.61±6.28%), much lower than that of the Mariana Trench sediments.
This difference may reflect different terrestrial influences since most marine samples in previous





studies are from continental margins where significant inputs of terrestrial-derived brGDGTs may
mask the marine signal (Hopmans et al., 2004).

Low BIT values (0.03±0.01; Fig. 6) in the Mariana Trench sediments is in similar to distal

marine sediments (an average of 0.04) (Schouten et al., 2013; Weijers et al., 2014), suggesting
insignificant terrestrial inputs. By compilation of globally distributed 1354 marine sediments and
589 soils, Xiao et al. (2016) found that the (IIIa+IIIa')/(IIa+IIa') ratio was < 0.59 in over 90% of
soils and 0.59–0.92 and > 0.92 in marine sediments with and without significant terrestrial inputs,
respectively. In this study, the (IIIa+IIIa')/(IIa+IIa') ratio varies between 5.68 and 8.33 (7.13±0.98)
(Fig. 6), much higher than the threshold value for marine origin (0.92), supporting in-situ production
of brGDGTs in the Mariana Trench sediments.

De Jonge et al. (2014a) proposed a CBT' index to reconstruct the soil pH based on global

distributed soils. Combined with new available data, we recalibrated the correlation of soil pH with
the CBT' index: $\text{pH}=(1.459 \pm 0.025) \times \text{CBT}'+(7.001 \pm 0.023)$ $(n = 628, \text{R}^2 = 0.84, p <$
$0.001)$ (Fig. 7a). According to this equation, the CBT' index of the Mariana Trench sediments
ranges from 0.78 to 0.90 (0.84±0.05) and the reconstructed pH is 8.22±0.07 (Fig. 7a). This pH is
very close to that of weak alkaline seawater (ca. 8.2), and therefore the brGDGTs in the Mariana
Trench are most likely produced in the marine environment.

Overall, the characters of bulk geochemical parameters, brGDGT compositions, the BIT index

and brGDGT-derived pH of soil and marine samples all support that brGDGTs in Mariana Trench
sediments are in-situ products rather than terrestrial inputs.

**4.2 High fractional abundance of brGDGT-IIIa' as a common phenomenon in marine**
**environments**

Not only Mariana Trench sediments, but also samples from continental margins show relatively

high proportions of hexamethylated 6-methyl brGDGTs. Dearing Crampton-flood et al. (2018)
explored brGDGTs and bulk properties of organic matter in a sediment record from the North Sea
Basin during the period of early Pliocene to early Pleistocene. The OC content, $\delta^{13}$C value, BIT and
#rings$_{\text{tetra}}$ index indicate a transition from predominant marine OC during the Pliocene to
predominant terrestrial OC in the Pleistocene. Correspondingly, the fractional abundance of





brGDGT-IIIa' is higher during the Pliocene (8.06±1.92%) than the Pleistocene (5.16±0.83%), and
exhibits significant correlations with $\delta^{13}$C ($R^2 = 0.68$, $p < 0.001$) and the BIT index ($R^2 = 0.46$, $p <$
0.001) (Fig. 8a, b, c). These correlations support that the variation in OC source controls the
composition of brGDGTs.

Another supporting evidence is from the Kara Sea. De Jonge et al. (2016) investigated

sedimentary brGDGT record of the Kara Sea spanning a minimum of 13.3 ka. The greater marine
OC contribution in the shallow sediments (1–130 cm; < 10 ka) was revealed by heavier $\delta^{13}$C (up to
–23‰) and lower BIT index (close to 0) compared to deep sediments (Fig. 8e, f). Coincide with this
change, the fractional abundance of GDGT-IIIa' appeared to be increasing from < 5% to 15% (Fig.
8d). Similar to the North Sea Basin, the significant correlations of the fractional abundance of
brGDGT-IIIa' with the $\delta^{13}$C ($R^2 = 0.34$; $p < 0.001$) and the BIT index ($R^2 = 0.50$; $p < 0.001$) were
observed in the Kara Sea, again suggesting that marine organisms tend to produce more
hexamethylated 6-methyl brGDGTs.

Besides temporal variations in sediment cores, the fractional abundance of 6-methyl brGDGTs

also varied spatially in modern samples from land to sea. Warden et al. (2016) examined brGDGTs
along a transect from the Tagus River into the deep ocean off the Portuguese margin. From source
to sink in the Tagus River basin, the BIT index decreases from 0.9 to < 0.1, reflecting a substantial
increase in marine contribution to sedimentary OC pool (Fig. 8h). Meanwhile, the proportion of
brGDGT-IIIa' increases from 11.07±2.62% to 29.31±6.45%, and brGDGT-IIIa' became the most
abundant compound in the Lower Setúbal Canyon sediments (Fig. 8g). Sinninghe Damsté (2016)
reported brGDGT composition in surface sediments from the Berau River delta including two coast-
shelf transects, and proposed #rings$_{tetra}$ index to discern sources of brGDGTs. The #rings$_{tetra}$ index
shows a marked increase from the river mouth (0.22) to the shelf break (0.83). By compiling the
data in Sinninghe Damsté (2016), we found that the proportion of brGDGT-IIIa' generally increases
seawards, presenting a similar distribution pattern as that of the $\delta^{13}$C and BIT index (Fig. 8i, j, k).
These spatial variations confirm that that marine in-situ production of brGDGTs is characterized by
the high fractional abundance of hexamethylated 6-methyl isomer.

In sum, the studies for the Kara Sea (De Jonge et al., 2016), the North Sea Basin (Dearing

Crampton-flood et al., 2018), the Tagus River basin (Warden et al., 2016) and the Berau River delta





(Sinninghe Damsté, 2016) all demonstrate increasing proportion of 6-methyl brGDGTs (particularly
IIIa') with intensified marine influence. These findings, along with the strong predominance of
brGDGT-IIIa' in the Mariana Trench sediments, suggest that the high proportion of brGDGT-IIIa'
is a common phenomenon in marine environments where in-situ production of brGDGTs is
significant.

**4.3 Potential mechanisms to produce high proportions of brGDGT-IIIa' in marine**
**environments**
A survey of brGDGTs in globally distributed soils suggests that brGDGT producing microbes
can adjust their membrane lipid compositions in response to environmental conditions, reflected by
the increase in cyclization degree of brGDGTs and the shift from 5- to 6-methyl group with
increasing pH and decreasing methylation of brGDGTs with temperature (Weijers et al., 2007b; De
Jonge et al., 2014a; Ding et al., 2015; Xiao et al., 2015). This adaption mechanism may be
extrapolated to marine organisms. In the Mariana Trench, in-situ production yields brGDGTs with
the strong predominance of 6-methyl (84.57±1.53%) (Table 2). The cyclopentane-containing
brGDGTs (Ib, Ic, IIb, IIb', IIc, IIc', IIIb, IIIb', IIIc, IIIc') comprise only 3.69±0.75% of total brGDGTs,
and the #rings$_{tetra}$ index is low (0.26±0.04). This seems contrast to a view that the fractional
abundance of cyclopentane-containing brGDGTs is positively correlated with pH (Sinninghe
Damsté, 2016). This discrepancy can be explained by two reasons. First, the isomerization of
brGDGTs, relative to the cyclization of brGDGTs, is a more effective way in response to changing
pH. Based on global soil dataset, the correlation between the Isomerization of Branched Tetraethers
index (IBT; Xiao et al., 2015) and pH is substantially higher ($n = 610$, $R^2 = 0.80$, $p < 0.001$, Fig. 7b)
than that between the #rings$_{tetra}$ index and pH ($n = 631$, $R^2 = 0.46$, $p < 0.001$, Fig. 7d) as well as that
between the cyclization index (CBT$_{5me}$) (De Jonge et al., 2014a) and pH ($n = 622$, $R^2 = 0.67$, $p <$
0.001, Fig. 7c). Meanwhile, the global soils with pH > 8 ($n = 58$) are characterized by higher
fractional abundance of 6-methyl brGDGTs (68.22±10.41%) than the cyclopentane-containing
brGDGTs (16.69±9.43%). The second explanation is that brGDGT producing microbes tend to
produce more hexamethylated brGDGTs at low temperature (Sinninghe Damsté, 2016), thus
reducing the relative proportion of cyclic tetramethylated and pentamethylated brGDGTs. Based on





the global dataset, if we take 100% of tetramethylated brGDGTs as a starting point, a decreasing
proportion of tetramethylated brGDGTs, most likely caused by decreasing temperature (Weijers et
al., 2007b), is initially compensated by a roughly linear increase of pentamethylated brGDGTs (Fig.
9d) and, to less extent, by a slower increase of hexamethylated brGDGTs (Fig. 9b). However, when
tetramethylated brGDGTs decreases to 20% of total brGDGTs, hexamethylated brGDGTs become
dominant, whereas pentamethylated brGDGTs reach a turning point and begin to rapidly decrease
(Fig. 9b, d). The ternary diagram, plotted with fractional abundance of tera-, penta- and hexa-
methylated brGDGTs (Fig. 9), shows that the Mariana Trench sediments have distinct and high
fractional abundance of hexamethylated brGDGTs (73.53±2.56%). We thus propose that low
temperature and high pH of deep-sea environments are responsible for production of brGDGTs with
high degree of methylation and the predominance of 6-methyl brGDGTs, especially brGDGT-IIIa'.
Marine in-situ production of brGDGTs may take place in water column, or sediments, or both.
Sinninghe Damsté (2016) suggested in-situ production of brGDGTs is a widespread phenomenon
in shelf sediments that is especially pronounced at water depths of ca. 50–300 m. Based on an
extended dataset of brGDGTs in open sea sediments (water depth 63–5521 m), the reconstructed
pH ranges from 6.1 to 9.9 (Weijers et al., 2014). This indicates that the brGDGTs was mainly
produced in benthic sediments where the pH of porewater is more variable than that of the water
column. However, the CBT'-derived pH in the Mariana Trench fell in a narrow range (8.22±0.07),
which are in line with the pH of the water column. Additionally, the brGDGTs-reconstructed
temperature using the $MAT_{mr}$ index ranged from 9.6 to 10.7 °C (10.2±0.3 °C), which is close to the
water temperature at ca. 300 m, but much higher than the temperature of benthic sediments (1.2 °C)
(Takuro et al., 2015; Tian et al., 2018). Given these facts, in-situ production of brGDGTs can occur
in both water column and benthic sediments, although the contribution weight of each sources may
be site-specific.

**4.4 Deciphering brGDGT provenance in marine sediments**
There are increasing concerns about the applicability of the brGDGT-based proxies in
continental margins which are characterized by intense land-sea interaction (De Jonge et al., 2016;
Sinninghe Damsté, 2016; Dearing Crampton-flood et al., 2018). Determining the provenance of



brGDGTs is prerequisite for accurate application of brGDGTs-based proxies. Our study highlights
that in-situ produced brGDGTs tend to exhibit higher fractional abundance of brGDGT-IIIa' relative
to terrestrial brGDGTs in most soil and peat samples. However, the fractional abundance of
brGDGT-IIIa' alone cannot decipher soil and marine source of brGDGTs since fractional abundance
of brGDGT-IIIa' in soils are variable and can reach up to 51% (De Jonge et al., 2014a). Xiao et al.
(2016) proposed the (IIIa+IIIa')/(IIa+IIa') ratio of < 0.59, 0.59–0.92 and > 0.92 to indicate an origin
of brGDGTs from soils, marine sediments with terrestrial influence and marine sediments without
terrestrial influence, respectively. However, the updated dataset shows some overlaps of the
(IIIa+IIIa')/(IIa+IIa') values between soils and marine sediments (Fig. 10). In order to circumvent
this problem, we propose a new approach to evaluate the source of brGDGTs based on the slope of
the (IIIa+IIIa')/(IIa+IIa') ratio and fractional abundance of brGDGT-IIIa' (Fig. 10). Specifically, the
slope of global soils (30.5±0.7) is substantially greater than that of marine sediments with terrestrial
influence (8.2±0.1), both of which are substantially greater than the slope of the Mariana Trench
sediments without terrestrial influence (2.3±0.3) (Fig. 10). The extremely low slope of Mariana
Trench sediments likely suggests that brGDGT are completely derived from in-situ production.
The systematic differences in the composition of brGDGTs between terrestrial and marine
production inevitably affect brGDGTs proxies. Since the CBT' index involves brGDGT-IIIa', the
marine in-situ production of brGDGTs with higher fractional abundance of brGDGT-IIIa' is very
likely to impact the CBT'-pH proxy. Although the brGDGT based temperature proxies, like
$MBT'_{5me}$ and $MAT_{mr}$, do not directly involve brGDGT-IIIa' (De Jonge et al., 2014a), in-situ
production of hexamethylated 6-methyl brGDGT will cause changes in proportions of tetra- and
pentamethylated brGDGTs to different degrees (Fig. 9b, c), and thereby influence brGDGTs based
temperature proxies.

**5.   Conclusions**
This work represents the first study for 5-methyl and 6-methyl brGDGT in sediments from the
Mariana Trench, the deepest ocean realm, from which we have reached three conclusions.
1)   The Mariana Trench sediments are characterized by the strong predominance of 6-methyl
brGDGTs (84.57±1.53% of total brGDGTs), especially brGDGT-IIIa' (73.40±2.39%), whereas 5-





methyl brGDGTs are below detection limit. This unique feature has never been previously reported
and is attributed to a combined effect of the lack of terrestrial input, alkaline seawater and low
subsurface temperature in the Mariana Trench.
2)      High (IIIa+IIIa')/(IIa+IIa') values (7.13±0.98), enriched $\delta^{13}C$ signatures (–19.82±0.25%),
low OC/TN ratios (6.72±0.84) and low BIT index (0.03±0.01) strongly suggest an in-situ production
of brGDGTs. By compiling brGDGT dataset from 634 soil, 473 peat, 88 river, 410 lake and 426
marine samples, we recalibrate the correlation of soil pH with the CBT' index ($R^2 = 0.84$, $p < 0.001$).
The reconstructed CBT'-pH (8.22±0.07) is close to weak alkaline seawater, while the $MBT_{mr}$
reconstructed temperature (10.2±0.3 °C) is close to water temperature at ca. 300 m deep, suggesting
a principal contribution of planktonic bacteria to the brGDGT pool in the Mariana Trench sediments.
3)      BrGDGTs in sediments from the Mariana Trench and several continental margins were
found to comprise higher fractional abundance of hexamethylated 6-methyl brGDGTs with
intensified marine influence. The slope of fractional abundance of brGDGT-IIIa' and the
(IIIa+IIIa')/(IIa+IIa') index can be used to decipher the terrestrial and marine provenance of
brGDGTs. Since in-situ production of predominant hexamethylated 6-methyl brGDGT influences
the robustness of brGDGT-based proxies, this study provides a new way to estimate brGDGT
sources and holds some promise in reducing uncertainty of brGDGTs-based paleoenvironmental
proxies.

**Data availability**: Data have been made available through FIGSHARE:
https://doi.org/10.6084/m9.figshare.9896120.v1 (Xiao et al., 2019)

*Acknowledgements* Samples were obtained during cruises by the RV Zhangjian. We are grateful to
captains, crews, and scientific personnel for their excellent support to obtain the samples. Weicheng
Cui, Binbin Pan and Jiasong Fang are thanked for their assistance in cruise organization, lander
design and instrumental analyses, respectively. The cruise is financially supported by the Shanghai
Committee of Science and Technology (15DZ1207000). The additional financial support is from
National Natural Science Foundation of China (41976030; 41676058) to Y. Xu.



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





**Table 1.** Organic carbon (OC) content, total nitrogen (TN) content, molar ratio of OC/TN and stable

carbon isotopic composition ($\delta^{13}$C) in the Mariana Trench sediments

| Sample ID | Depth (cm) | OC (wt. %) | TN (wt. %) | OC/TN (mol/mol) | $\delta^{13}$C (‰) |
|---|---|---|---|---|---|
| MT1 | 0-2 | 0.31 | 0.05 | 6.52 | −20.02 |
| MT2.5 | 2-3 | 0.27 | 0.05 | 6.05 | −19.66 |
| MT3.5 | 3-4 | 0.29 | 0.05 | 6.85 | −19.55 |
| MT4.5 | 4-5 | 0.27 | 0.05 | 5.78 | −19.84 |
| MT5.5 | 5-6 | 0.29 | 0.06 | 6.13 | −19.94 |
| MT6.5 | 6-7 | 0.30 | 0.06 | 5.62 | −19.47 |
| MT7.5 | 7-8 | 0.27 | 0.04 | 7.27 | -19.54 |
| MT8.5 | 8-9 | 0.29 | 0.05 | 6.93 | −19.82 |
| MT9.5 | 9-10 | 0.28 | 0.04 | 7.74 | −20.09 |
| MT10.5 | 10-11 | 0.26 | 0.04 | 8.34 | −20.27 |





**Table 2.** Fractional abundance and concentration of brGDGTs and crenarchaeol (cren) in the Mariana Trench sediments.

| Sample ID | Ia (%) | Ib (%) | Ic (%) | IIa (%) | IIa' (%) | IIb (%) | IIb' (%) | IIc (%) | IIc' (%) | IIIa (%) | IIIa' (%) | IIIb (%) | IIIb' (%) | IIIc (%) | IIIc' (%) | BrGDGTs (ng/g) | Cren (ng/g) |
|---|---|---|---|---|---|---|---|---|---|---|---|---|---|---|---|---|---|
| MT1 | 13.6 | 2.7 | 1.5 | 0.0 | 10.4 | 0.0 | 0.0 | 0.0 | 0.0 | 0.0 | 71.8 | 0.0 | 0.0 | 0.0 | 0.0 | 18.4 | 353.3 |
| MT2.5 | 13.5 | 2.4 | 1.6 | 0.0 | 12.1 | 0.0 | 1.3 | 0.0 | 0.0 | 0.0 | 69.0 | 0.0 | 0.0 | 0.0 | 0.0 | 14.7 | 426.7 |
| MT3.5 | 11.1 | 1.4 | 0.6 | 0.0 | 9.5 | 0.0 | 0.6 | 0.0 | 0.0 | 0.0 | 76.2 | 0.0 | 0.6 | 0.0 | 0.0 | 16.4 | 659.8 |
| MT4.5 | 14.2 | 1.4 | 0.9 | 0.0 | 9.2 | 0.0 | 0.4 | 0.0 | 0.0 | 0.0 | 73.9 | 0.0 | 0.0 | 0.0 | 0.0 | 12.6 | 515.4 |
| MT5.5 | 11.1 | 2.0 | 0.8 | 0.0 | 10.3 | 0.0 | 0.8 | 0.0 | 0.0 | 0.0 | 75.0 | 0.0 | 0.0 | 0.0 | 0.0 | 15.1 | 622.7 |
| MT6.5 | 11.2 | 2.1 | 0.9 | 0.0 | 9.1 | 0.0 | 0.0 | 0.0 | 0.0 | 0.0 | 76.0 | 0.0 | 0.8 | 0.0 | 0.0 | 20.1 | 667.2 |
| MT7.5 | 13.4 | 1.5 | 1.0 | 0.0 | 11.3 | 0.0 | 1.2 | 0.0 | 0.0 | 0.0 | 71.5 | 0.0 | 0.0 | 0.0 | 0.0 | 12.7 | 551.8 |
| MT8.5 | 13.0 | 2.2 | 1.1 | 0.0 | 12.7 | 0.0 | 0.0 | 0.0 | 0.0 | 0.0 | 70.9 | 0.0 | 0.0 | 0.0 | 0.0 | 13.0 | 585.1 |
| MT9.5 | 11.8 | 2.0 | 0.7 | 0.0 | 9.2 | 0.0 | 0.0 | 0.0 | 0.0 | 0.0 | 76.3 | 0.0 | 0.0 | 0.0 | 0.0 | 11.5 | 450.6 |
| MT10.5 | 11.8 | 1.8 | 1.0 | 0.0 | 10.6 | 0.0 | 1.0 | 0.0 | 0.4 | 0.0 | 73.3 | 0.0 | 0.0 | 0.0 | 0.0 | 14.3 | 498.3 |




**Figure 1.** Location of the samples in this study. Red, blue, green, orange and pink circles indicate
globally distributed soil, river, lake, peat and marine samples, respectively. Black star denotes the
sediment core in the Mariana Trench. The detailed information is provided in supplementary
material.

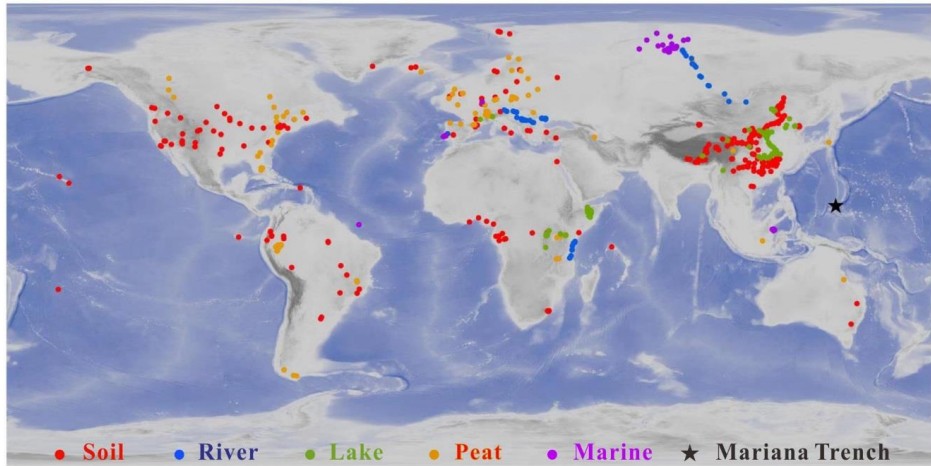




**Figure 2.** Chemical structures of brGDGTs and crenarchaeol.

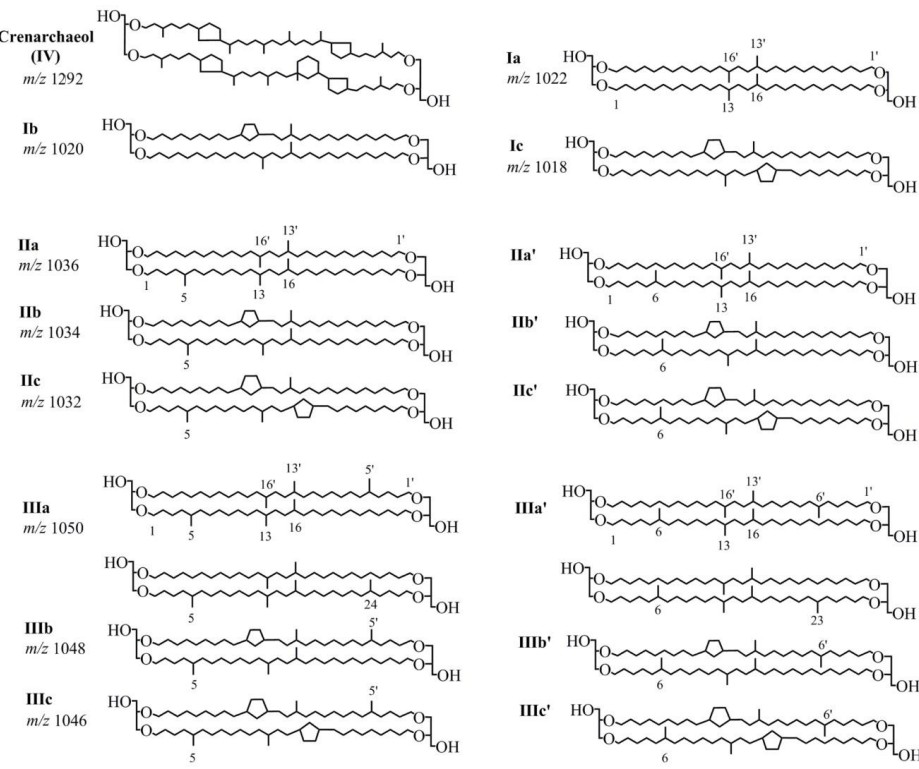

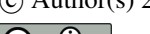



**Figure 3.** Extracted ion chromatograms (EICs) of m/z 1050 (left) and m/z 1036 (right) showing separation of 5-methyl and 6-methyl brGDGTs in soil (top), Mariana Trench sediment (middle) and combined soil and sediment (bottom).

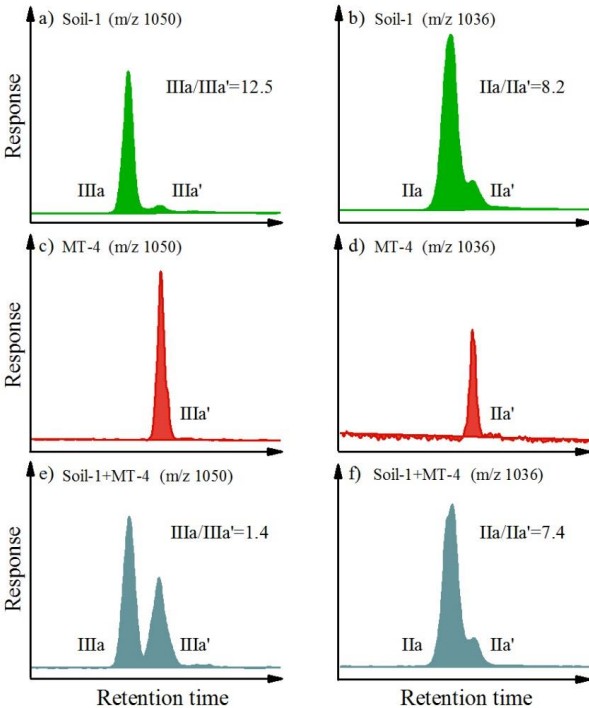




**Figure 4.** Plot of $\delta^{13}C$ versus TN/OC for core sediments from the Mariana Trench (MT). Included
in this graph are different compositional ranges of $C_3$ vascular plants, $C_4$ vascular plants, bacteria,
river and estuary phytoplankton and marine phytoplankton sources. The compositional range of
different end members was cited from Goñi et al. (2006) and Hu et al. (2016). The red stars and
green stars denote data from this study and Luo et al. (2017), respectively.

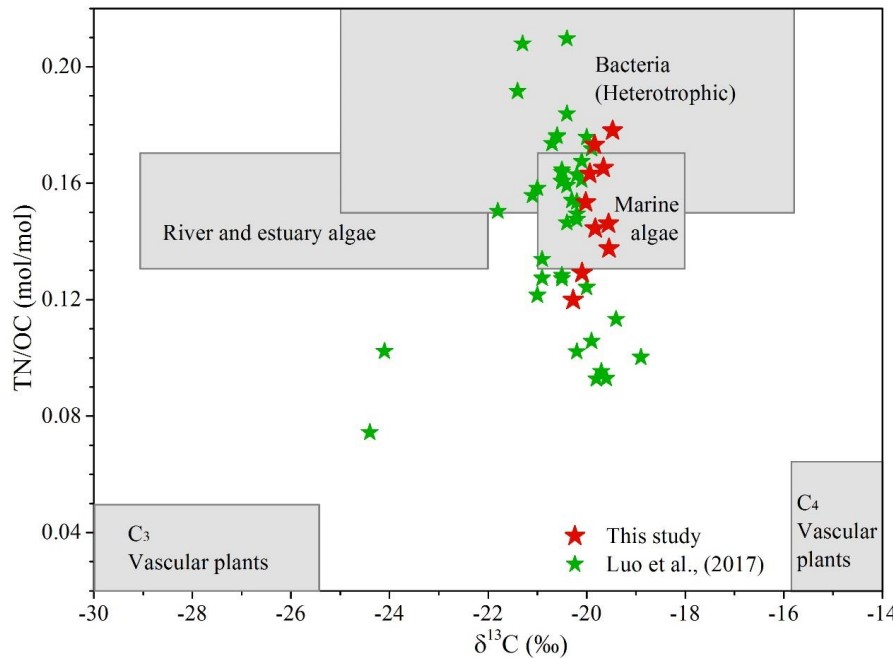








**Figure 5.** Comparisons of distribution of 15 brGDGT compounds in soil (n = 634), peat (n = 473),

river (n = 88), lake (n = 410), marine (n = 415) and Mariana Trench (n = 11) samples.

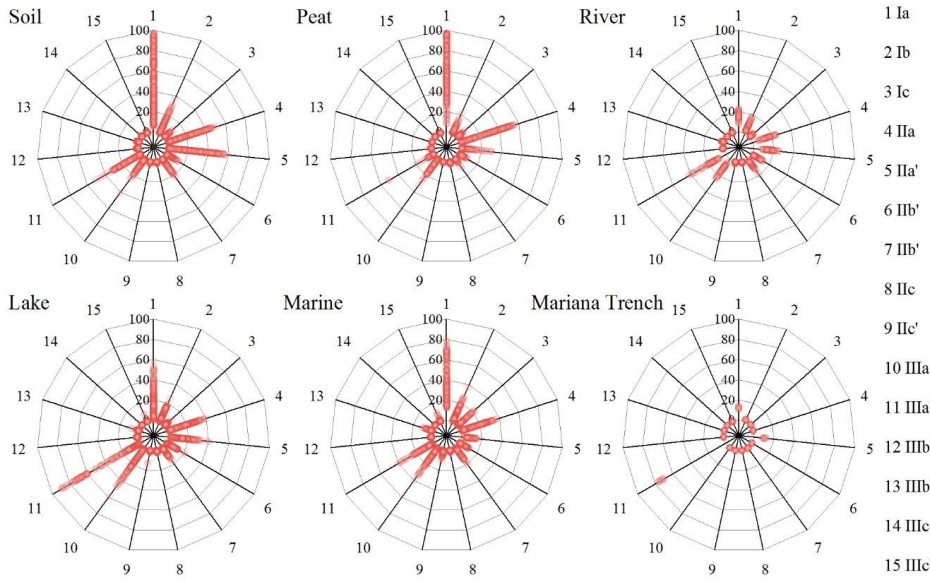






**Figure 6.** Relationship between the (IIIa+IIIa')/(IIa+IIa') index and the BIT index of the Mariana

Trench sediments (red star) and globally distribute soil (green circle) and marine samples (blue

square). The dashed lines represent the upper limit of production in the terrestrial realm and the

lower limit of production in the marine realm defined by Xiao et al. (2016).

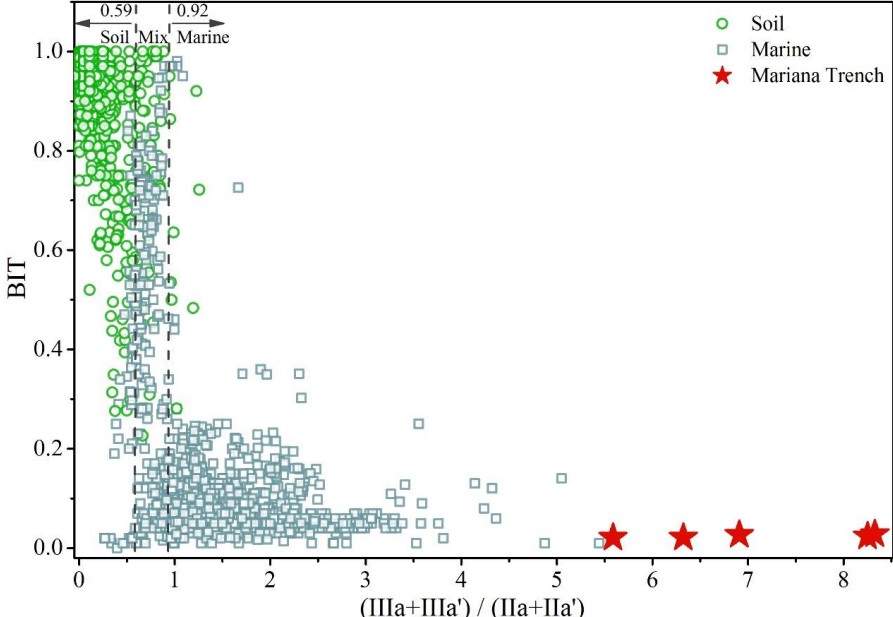



**Figure 7.** Scatterplots of the a) CBT', b) IBT, c) CBT$_{5me}$ and d) #rings$_{tetra}$ index versus measured

pH of globally distributed soils. The black solid line and dashed line denote the linear calibration

line and associated confidence intervals of 95%. The gray block (a) represents corresponding values

of the Mariana Trench sediment in this study.

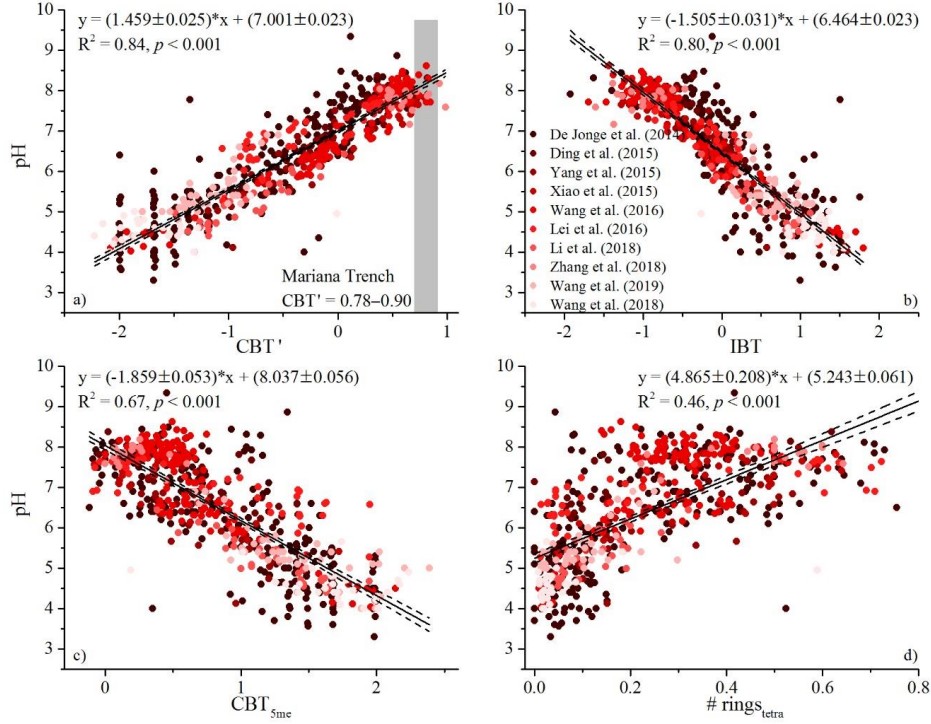








**Figure 8.** Vertical profiles of a) the proportion of GDGT-IIIa', b) $\delta^{13}C$ and c) BIT index of a marine
sediment core from the North Sea Basin (Dearing Crampton-flood et al., 2018). Vertical profiles of
d) the proportion of GDGT-IIIa', e) $\delta^{13}C$, f) BIT index of a marine sediment core from the Kara Sea
(De Jonge et al., 2016). Spatial distribution patterns of g) average distribution of brGDGTs and h)
BIT index in the transect from the land to the ocean off the Portuguese coast (river floodplain,
mudbelt, Lisbon canyon head and lower Setúbal canyon) (Warden et al., 2016). Isosurface plots of
i) BIT index, j) $\delta^{13}C$ and k) the proportion of GDGT-IIIa' of the surface sediments from the Berau
River delta (Sinninghe Damsté, 2016).

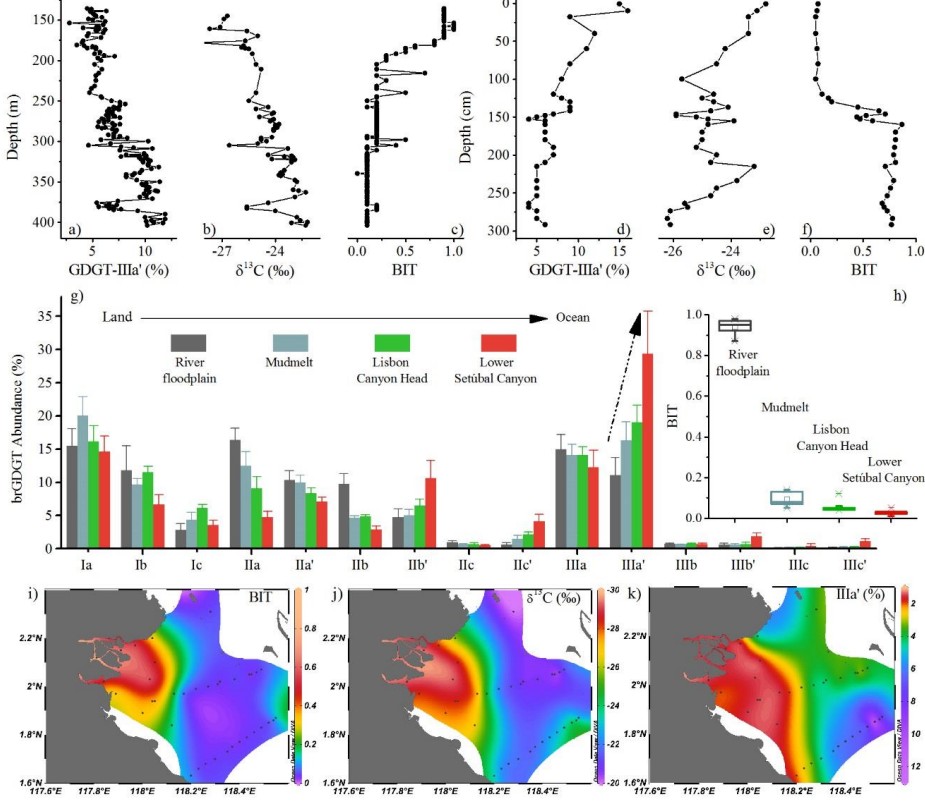






**Figure 9** a) Ternary diagram showing the fractional abundances of tetra-, penta- and hexamethylated
brGDGTs. b) and d) Cross plots of the fractional abundances of tetramethylated brGDGTs versus
hexa- and pentamethylated brGDGTs, respectively. c) Cross plots of the fractional abundances of
pentamethylated brGDGTs versus hexamethylated brGDGTs. The compiled dataset (Supplementary)
includes globally distributed soil, peat, lake, river and marine samples, as well as the Mariana Trench
sediments.

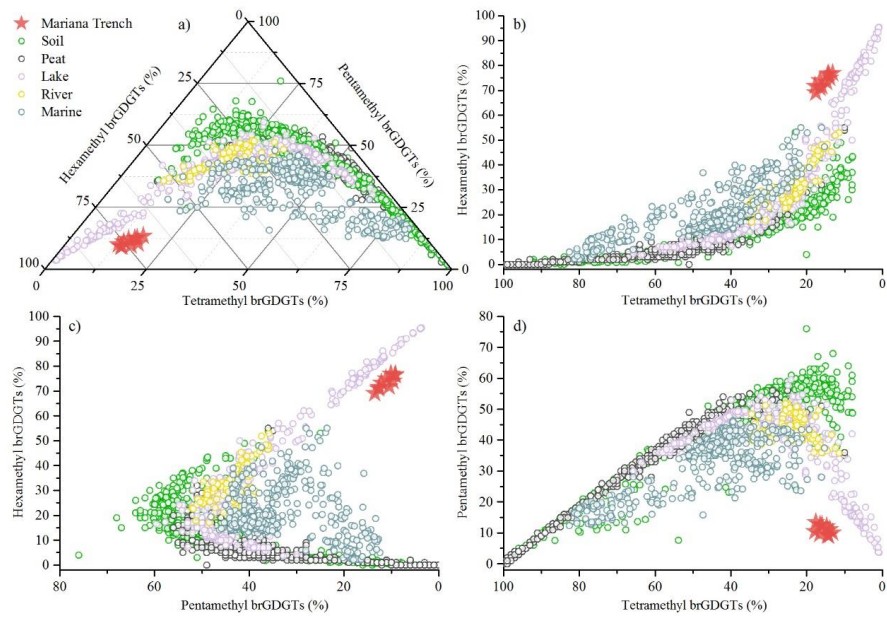





**Figure 10** Scatterplots of the (IIIa+IIIa')/(IIa+IIa') index versus the proportion of brGDGT-IIIa' of

globally distributed soils and marine sediments. The solid, dashed and dotted line denotes the Linear

fit, 95% confidence band and 95% prediction band of concatenated data, respectively. The number

of samples, slope, $R^2$ and $p$ values of calibration for the global distributed soils, marine sediments

and Mariana Trench sediments are given.

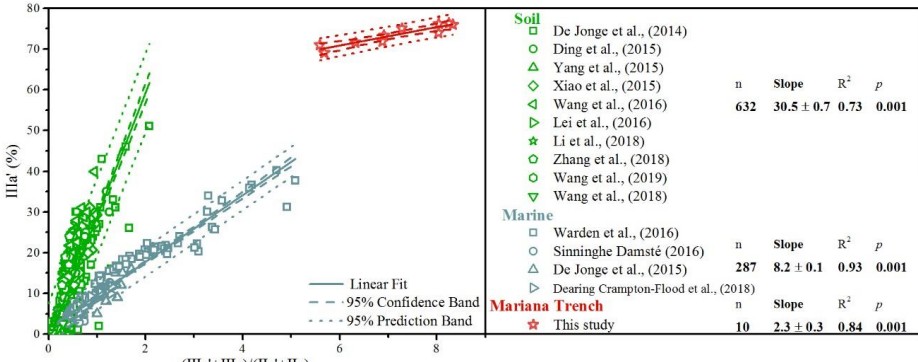