# Peer review of "Predominance of hexamethylated 6-methyl branched glycerol dialkyl glycerol"

_Biogeosciences, 2019_

## Referee Comment (RC1) · Anonymous Referee #1 · 31 Oct 2019

The manuscript by Xiao et al. presents a very interesting and welcome dataset on the distribution of brGDGTs in the Mariana Trench sediments. The results show distinct predominance of hexamethylated 6-methyl brGDGTs in this deepest ocean, suggesting brGDGTs are produced in situ by indigenous bacteria. This provides new insights into the composition of brGDGTs in "uncontaminated" marine and its usefulness as an endmember for brGDGTs source trace. It is suitable for Biogeosciences, but I have some suggestions/comments that could possibly improve the manuscript.

General Comments:

1. The current introduction dwells heavily on the history of brGDGTs, addressing their application as biomarkers for paleoclimate reconstruction, but falls short on giving much insight into the microbial ecology of this mysterious marine system. Oxygen concentration almost certainly plays a strong role in structuring the GDGT-producing microbial communities. Most brGDGTs were found produced by anaerobic bacteria. I strongly recommend the authors devote more space in the manuscript to discussing their data in the hydrologic and biogeochemical context of the MT.

2. I think that the paper would benefit from less discussion about the global brGDGT index application, which was already deeply discussed in Xiao et al. (2016). It dilutes the main conclusion of the paper, which is the distinct predominance of hexamethylated 6-methyl brGDGTs but also the absence of 5-methyl brGDGTs in the deepest ocean, this is an exciting result and again I hope the authors could devote more space on the explanations. Although some papers have already reported the dominance of hexamethylate brGDGTs in the marine sediments, none of them found the absence of 5-methyl brGDGTs. To me, it should be mainly driven by specific producers in this extreme environment than the environmental condition changes.

3. The application of soil pH index CBT' and mean annual air temperature index MATmr in this marine setting is unconvincing. To be noticed, both CBT' and MATmr were established using stepwise forward selection, see De jonge et al. (2014) and Loomis et al. (2012), which are only suitable for terrestrial regions and have no mechanism behind compare to CBT/IBT or MBT.

4. Results and discussion not completely separated There is some overlap here, with data appearing in the discussion.

5. The cross plot of acyclic hexa-/pentamethylated brGDGTs ratio and fractional abundance of brGDGT-IIIa' as a new approach to distinguish the terrestrial vs. marine provenance of brGDGTs (Fig. 10) can be removed. As the relationship between the (IIIa+IIIa')/(IIa+IIa') index and the BIT index (Fig. 6; Xiao et al., 2016) have already

clearly separated the Mariana Trench sediments from the other samples. I think that it could be eliminated with no loss to the manuscript (If it remains, its value should be made more clear, why it's important compare to the index set before).

6. There is an excessive number of figures, leading to some redundancy. I would delete some overlapping ones and focus more on the main point (see specific points of clarification below).

Specific comments:

Line 15: Leave space between numbers and symbols, keep consistent format for the left manuscript. Keep one decimal place for the relative abundance of brGDGTs for the following parts.

Line 22-24: See my general comments

Line 36: Change the literature order in the reference. The early shown one should be named Weijers et al. (2007a)

Line 43-45: One sentence missing: While isoGDGTs were mainly produced in the marine realm. Thus. . .

Line 57: Schouten et al. (2007) is misused here.

Line 60: brGDGTs with cyclopentanes are not called hexa- or pentamethylated brGDGTs. Please correct this sentence.

Line 111: Please cite Huguet et al. (2006) here.

Line 112: (3:1, v/v)

Line 122: Italic m/z, correct followings.

Line 195:A description of HPLC/MS method in the Method Section is missing. Either add in or cite a paper. If this is not a method developed by the authors, please rephrase the sentence.

Line 197: "...and hexamethylated (m/z 1050) brGDGTs in sediments of the Mariana Trench. This feature shows a distinct difference..."

Line 248-253: This part should go to results.

Line 242-263: As the authors mentioned two times in both introduction and method, this deepest trench is remote from any mainland and has no significant terrestrial influence. Either shorten it using one or two sentences or delete it.

Line 264: ...is similar to those of distal marine sediments... Schouten et al. (2013) is not a good reference the way it is written here.

Line 266: Recheck the numbers you used here. 1354 soils and 589 marine sediments should be.

Line 272-278: see my general comments

Section 4.2: Would it be nice to add in the data of this study for comparison, since all of the data the authors choose are sea sediments.

Line 333-336: Again results.

Line 341-346: This is more a comparison of the indexes than the discussion of mechanisms behind. I would suggest to delete it. Only talking about the reasons for the predominance of IIIa' here, 1) sedimentary in-situ brGDGTs-producers produce more hexamethylated brGDGTs to adapt to the low temperature/poor nutrient conditions (Sinninghe Damsté, 2016; Ding et al., 2018); 2) brGDGTs-producers adopt a strategy of the carbon skeleton isomerization of brGDGTs to live at alkaline seawater, resulting in a distribution in which 6-methyl brGDGTs are abundant (Ding et al., 2015; Xiao et al., 2015).

Line 349-372: See my general comments, I would condense this part and delete the discussion that dilutes the main findings.

Line 435: Recheck the format of references. Some doi can not be opened.

[Figure]

Fig. 2: Set a boundary between brGDGTs from Crenarchaeol, most external readers will not understand.

Fig. 7: Not needed, see the general comments above

Fig. 9: Delete Fig. b, c and d since they are showing the same results as a.

---

## Referee Comment (RC2) · Anonymous Referee #2 · 30 Dec 2019

The manuscript by Xiao et al. describes the distribution of branched tetraether lipids in Mariana Trench sediments. Very few studies to date have investigated the organic geochemistry of deep-sea trench sediments and even fewer have studied the distribution of tetraether lipids. The study presented by Xiao et al. thus is a novel contribution to the field. Specifically, the authors show that the distribution of branched tetraethers is unique when compared to previously studied environments.

I have two major criticisms, or suggestions, that the authors should consider in preparing a revised version.

1) The uniqueness of the study site is both a strength and a weakness of the presented work. It is a strength, as the remote setting may allow distinguishing marine in situ production from a terrestrial origin of brGDGTs that muddles interpretation of shelf sediments. However, it is a weakness because it is unclear how comparable the site is to continental shelf sediments. This is regardless of whether brGDGTs originate from sediments or from the water column, since factors such as nutrients, particle loading, bacterial community composition, oceanographic parameters (oxygenation, salinity, currents etc.) will vary between the shelf and trench sediments and between shelf water column and the pelagic water column above the trench. These points are particularly important since it remains unresolved whether brGDGT production in the ocean originates from the water column or sediments, or both, and which bacterial clades synthesize brGDGTs. The authors should address these caveats in their manuscript. A good place to discuss these issues would be between lines 374-400. This discussion should then be reflected in the revised abstract.

2) I fundamentally disagree with the use of soil calibrations for reconstructing seawater, or porewater, pH and temperature (lines 360-372). There is no evidence supporting the applicability of these calibrations. Therefore, the brGDGT-based pH reconstructions cannot be used as evidence for in situ production. Specifically, it is currently unclear if the clades of bacteria producing brGDGTs in soils are similar to those in the ocean, particularly because the perceived adaptation of branched GDGT distributions to environmental parameters has previously been suggested to be a community effect.

I think that the number of figures in the main text could be reduced in order to streamline the manuscript. I suggest moving figures 3-5 to the supplement.

Finally, I recommend the authors carefully examine their manuscript to fix multiple typos and grammar.

In summary, I recommend publication after moderate changes have been made.

Additional comments:

[Figure]

All figures: To ensure accessibility, please use color blind-friendly colors, e.g., do not use red and green in the same figure.

Line 8-24: Please correct grammar issues throughout the abstract

Line 16: Please specify that d13C values are for OC

Line 28-30: Please consider finding more appropriate citations. Sinninghe Damste 2000 and Weijers et al. are neither the first to report on iGDGTs/brGDGTs nor are these the most comprehensive papers.

Line 120-122: Without response factors of GDGTs relative to the C46 standard, concentrations cannot be determined. Please report concentrations as response units or peak areas normalized to OC.

---

## Author Comment (AC1) · 19 Jan 2020

The manuscript by Xiao et al. presents a very interesting and welcome dataset on the distribution of brGDGTs in the Mariana Trench sediments. The results show distinct

predominance of hexamethylated 6-methyl brGDGTs in this deepest ocean, suggesting brGDGTs are produced in situ by indigenous bacteria. This provides new insights into the composition of brGDGTs in "uncontaminated" marine and its usefulness as an endmember for brGDGTs source trace. It is suitable for Biogeosciences, but I have some suggestions/comments that could possibly improve the manuscript. Response: We appreciate the reviewer 1 to acknowledge the merit of our work. He/She also provided valuable suggestion to improve our manuscript. The follows are our response to each comment.

General Comments: 1. The current introduction dwells heavily on the history of brGDGTs, addressing their application as biomarkers for paleoclimate reconstruction, but falls short on giving much insight into the microbial ecology of this mysterious marine system. Oxygen concentration almost certainly plays a strong role in structuring the GDGT-producing microbial communities. Most brGDGTs were found produced by anaerobic bacteria. I strongly recommend the authors devote more space in the manuscript to discussing their data in the hydrologic and biogeochemical context of the MT.

Response: This is a good comment. Currently, there are two main themes of GDGT studies. 1) Calibration and application of GDGTs-derived proxies as temperature, pH and OC source indicators. 2) Mechanism of GDGT biosynthesis by microbes (archaea or certain bacteria) using molecular biology techniques. In the original submission, we paid more attention to the first theme. In the revised manuscript, we added the contents about microbial ecology of GDGTs in introduction part. For example, from line 62-66, we wrote as "In addition, oxygen (Qin et al., 2015) and moisture (Dang et al., 2016a) was found to play a profound role in GDGT-biosynthesis besides temperature and pH. By examining vertical patterns of brGDGTs and bacterial 16S rRNA gene in a deep meromictic Swiss lake (Lake Lugano), Weber et al. (2018) suggest that brGDGTs are synthesized by multiple groups of bacteria thriving under contrasting redox regimes." We also cited these references. In the session 4.3, we added the discussion about

biogeochemical context in the Mariana Trench. From line 635 to 642, we wrote as "However, it should be pointed out that the bottom of Mariana Trench has the hydrostatic pressure > 100 MPa and is overlain by oligotrophic water masses with surface primary productivity of ca. 50 g OC m-2 yr-1 (Jamieson, 2015). Consequently, the unique microbes have been evolved in this extreme environment, such as proliferation of hydrocarbon-degrading bacteria (Liu et al., 2019), that may response to temperature and pH in a different way as their counterparts dwelling in shallow water regions. Nevertheless, the investigation of microbial community and intact polar lipids in the Mariana Trench is needed for understanding the source and environmental implication of brGDGTs in the deepest ocean." Please see our revised manuscript for the details.

2. I think that the paper would benefit from less discussion about the global brGDGT index application, which was already deeply discussed in Xiao et al. (2016). It dilutes the main conclusion of the paper, which is the distinct predominance of hexamethylated 6-methyl brGDGTs but also the absence of 5-methyl brGDGTs in the deepest ocean, this is an exciting result and again I hope the authors could devote more space on the explanations. Although some papers have already reported the dominance of hexamethylate brGDGTs in the marine sediments, none of them found the absence of 5-methyl brGDGTs. To me, it should be mainly driven by specific producers in this extreme environment than the environmental condition changes.

Response: We accepted this suggestion. BrGDGTs-derived proxies such as MBT, CBT and IBT are all developed based on terrestrial samples, and their correlations with environmental factors (e.g., temperature, pH) may be not suitable in marine settings. Thus, in the revised manuscript, we removed relevant contents about global application and calibration of brGDGTs. Specifically, we removed the figure 7 about a) CBT', b) IBT, c) CBT5me and d) #ringstetra index versus measured pH of globally distributed soils. We also deleted figs. 9b, c and d about global distributions of brGDGTs. The result and discussion about correlation between brGDGTs and environmental factors (temperature and pH) were also removed from the main text (e.g., the second

paragraph of session 4.4). Overall, in the revised manuscript, we emphasize the point about the predominance of 6-methyl brGDGTs and the absence of 5-methyl brGDGTs in the Mariana Trench.

3. The application of soil pH index CBT' and mean annual air temperature index MATmr in this marine setting is unconvincing. To be noticed, both CBT' and MATmr were established using stepwise forward selection, see De jonge et al. (2014) and Loomis et al. (2012), which are only suitable for terrestrial regions and have no mechanism behind compare to CBT/IBT or MBT.

Response: As mentioned above, we accepted the reviewer's suggestion and removed all relevant figures and discussion. It is true that brGDGTs-derived proxies such as MBT, CBT and IBT are all based on soil dataset, and could not be used for marine sediments directly if these compounds are produced by marine organisms. We keep this issue in our mind when revised the manuscript.

4. Results and discussion not completely separated. There is some overlap here, with data appearing in the discussion.

Response: We partly accepted this suggestion. We tried our best to move some unnecessary data presentation into the result part. However, the discussion can be benefited by brief data presentation.

5. The cross plot of acyclic hexa-/pentamethylated brGDGTs ratio and fractional abundance of brGDGT-IIIa' as a new approach to distinguish the terrestrial vs. marine provenance of brGDGTs (Fig. 10) can be removed. As the relationship between the (IIIa+IIIa')/(IIa+IIa') index and the BIT index (Fig. 6; Xiao et al., 2016) have already clearly separated the Mariana Trench sediments from the other samples. I think that it could be eliminated with no loss to the manuscript (If it remains, its value should be made clearer, why it's important compare to the index set before).

Response: We think the reviewer did not catch the point why we proposed the new

source indicator for brGDGTs in marine settings. It is true that Xiao et al. (2016) already proposed the (IIIa+IIIa')/(IIa+IIa') index to distinguish source of brGDGTs (soil vs. marine with and without terrigenous influence). However, there is still overlap for the (IIIa + IIIa')/(IIa + IIa') values between soils and marine sediments, as shown in Fig. 6. So, it is necessary to develop more sensitive indicator. In our study, we found the combination of the (IIIa + IIIa')/(IIa + IIa') ratio and fractional abundance of brGDGT-IIIa' can completely separate samples with different terrigenous influence. The cross-plot of these two indicators results in distinct difference in the slope among soils, marine sediments with different terrigenous influence (please see fig. 9). Thus, our new approach provides two dimensional resolution to assess source of brGDGTs, whereas the (IIIa + IIIa')/(IIa + IIa') ratio by Xiao et al. (2016) is only one dimensional resolution. Furthermore, we found the slope of the (IIIa + IIIa')/(IIa + IIa') ratio and fractional abundance of brGDGT-IIIa' is applicable for sediment cores by compiling literature data. Given these facts, we keep figure 9 and discussion about our new approach to distinguish source of brGDGTs (Session 4.4: Deciphering brGDGT provenance in marine sediments) in the revised manuscript.

6. There is an excessive number of figures, leading to some redundancy. I would delete some overlapping ones and focus more on the main point (see specific points of clarification below).

Response: We accepted this suggestion and deleted the fig.7, 9b, 9c, and 9d in the revised manuscript.

Specific comments: Line 15: Leave space between numbers and symbols, keep consistent format for the left manuscript. Keep one decimal place for the relative abundance of brGDGTs for the following parts.

Response: We accepted this suggestion and added the space between number and unit throughout the manuscript.

Line 22-24: See my general comments

Response: Please see our responses to the fifth general comment.

Line 36: Change the literature order in the reference. The early shown one should be named Weijers et al. (2007a)

Response: We made this correction in the revised manuscript.

Line 43-45: One sentence missing: While isoGDGTs were mainly produced in the marine realm. Thus...

Response: We added the sentence as "while iGDGTs are mainly produced in the marine realm" in line 52 of the revised manuscript.

Line 57: Schouten et al. (2007) is misused here.

Response: we removed Schouten et al. (2007) here.

Line 60: brGDGTs with cyclopentanes are not called hexa- or pentamethylated brGDGTs. Please correct this sentence.

Response: we accepted this suggestion and removed "hexa- or pentamethylated brGDGTs" here.

Line 111: Please cite Huguet et al. (2006) here.

Response: we accepted this suggestion and cited Huguet et al. (2006) in the revised manuscript.

Line 112: (3:1, v/v)

Response: Done.

Line 122: Italic m/z, correct followings.

Response: we accepted this suggestion and made correction throughout the manuscript.

Line 195: A description of HPLC/MS method in the Method Section is missing. Either

add in or cite a paper. If this is not a method developed by the authors, please rephrase the sentence.

Response: we accepted this suggestion. In the revised manuscript, we added a sentence as "The detailed instrumental parameters were described in Hopmans et al. (2016)."

Line 197: "...and hexamethylated (m/z 1050) brGDGTs in sediments of the Mariana Trench. This feature shows a distinct difference..."

Response: We accepted this suggestion. In the revised manuscript, we rewrote as "This feature is distinctly different from previous studies for other environmental settings that two or more peaks (5-methyl, 6-methyl and even 7-methyl isomers) were identified".

Line 248-253: This part should go to results.

Response: we did not accept this suggestion. Because here we discuss the difference of brGDGT compositions between Mariana Trench and other globally distributed samples. We think it is needed to supply some summarized data about individual brGDGTs, so the readers can easily catch this point. In addition, we only used one sentence to describe the difference. Considering these facts, we still keep this sentence in the discussion part.

Line 242-263: As the authors mentioned two times in both introduction and method, this deepest trench is remote from any mainland and has no significant terrestrial influence. Either shorten it using one or two sentences or delete it.

Response: We accepted this suggestion, and shorten this as one sentence "This difference may reflect a difference in terrestrial influence since most marine samples in literatures are from continental margins where significant contribution of terrestrial-derived brGDGTs may mask the marine signal." (line 468-471).

Line 264: ...is similar to those of distal marine sediments... Schouten et al. (2013) is

not a good reference the way it is written here.

Response: We accepted this suggestion and removed the reference of Schouten et al.(2013) here.

Line 266: Recheck the numbers you used here. 1354 soils and 589 marine sediments should be.

Response: We have corrected this mistake, and changed the numbers into "By compilation of globally distributed 1354 soils and 589 marine sediments"

Line 272-278: see my general comments. Section 4.2: Would it be nice to add in the data of this study for comparison, since all of the data the authors choose are sea sediments.

Response: Please see our responses to the general comments #2 and #3 above.

Line 333-336: Again results.

Response: we accepted this suggestion and deleted the data here. In the revised manuscript, we wrote the sentences as "This adaption mechanism may be extrapolated to marine organisms. In the Mariana Trench, in-situ production yields brGDGTs with the strong predominance of 6-methyl. The cyclopentane-containing brGDGTs (Ib, Ic, IIb, IIb', IIc, IIc', IIIb, IIIb', IIIc, IIIc') comprise less than 10% of total brGDGTs, and the #ringstetra index is low (Table 2)".

Line 341-346: This is more a comparison of the indexes than the discussion of mechanisms behind. I would suggest to delete it. Only talking about the reasons for the predominance of IIIa' here, 1) sedimentary in-situ brGDGTs-producers produce more hexamethylated brGDGTs to adapt to the low temperature/poor nutrient conditions (Sinninghe Damsté, 2016; Ding et al., 2018); 2) brGDGTs-producers adopt a strategy of the carbon skeleton isomerization of brGDGTs to live at alkaline seawater, resulting in a distribution in which 6-methyl brGDGTs are abundant (Ding et al., 2015; Xiao et al., 2015).

Response: We agree with the reviewer on the explanation of adaption mechanism of brGDGTs-producers to environmental factors. In the revised manuscript, we emphasize the unique feature of brGDGTs in the Mariana Trench and tried to explain its potential reason in context of the extremely environmental condition in this deepest ocean. We also deleted all contents about calibration of brGDGT parameters at the global scales, such as removal of figure 7, 9b,c and d as well as relevant contents in main text.

Line 349-372: See my general comments, I would condense this part and delete the discussion that dilutes the main findings.

Response: Please see our responses to the general comments above.

Line 435: Recheck the format of references. Some DOI cannot be opened.

Response: we already double checked our references and format in the revised manuscript.

Fig. 2: Set a boundary between brGDGTs from Crenarchaeol, most external readers will not understand.

Response: We have highlighted Crenarchaeol with a rectangular.

Fig. 7: Not needed, see the general comments above

Response: We accepted this suggestion and deleted Fig.7.

Fig. 9: Delete Fig. b, c and d since they are showing the same results as a.

Response: We accepted this suggestion and removed Fig. 9b, c, and d.

---

## Author Comment (AC2) · 19 Jan 2020

The manuscript by Xiao et al. describes the distribution of branched tetraether lipids in Mariana Trench sediments. Very few studies to date have investigated the organic

geochemistry of deep-sea trench sediments and even fewer have studied the distribution of tetraether lipids. The study presented by Xiao et al. thus is a novel contribution to the field. Specifically, the authors show that the distribution of branched tetraethers is unique when compared to previously studied environments. I have two major criticisms, or suggestions, that the authors should consider in preparing a revised version.

1) The uniqueness of the study site is both a strength and a weakness of the presented work. It is a strength, as the remote setting may allow distinguishing marine in situ production from a terrestrial origin of brGDGTs that muddles interpretation of shelf sediments. However, it is a weakness because it is unclear how comparable the site is to continental shelf sediments. This is regardless of whether brGDGTs originate from sediments or from the water column, since factors such as nutrients, particle loading, bacterial community composition, oceanographic parameters (oxygenation, salinity, currents etc.) will vary between the shelf and trench sediments and between shelf water column and the pelagic water column above the trench. These points are particularly important since it remains unresolved whether brGDGT production in the ocean originates from the water column or sediments, or both, and which bacterial clades synthesize brGDGTs. The authors should address these caveats in their manuscript. A good place to discuss these issues would be between lines 374-400. This discussion should then be reflected in the revised abstract.

Response: This is a good suggestion. Although we have mentioned this point, we did not explain it systematically. So, in the revised manuscript, we accepted the reviewer's suggestion and added this content in session 4.3. We added a paragraph as "The unique feature in the composition of brGDGTs in the Mariana Trench has significant implications on the brGDGTs-derived proxies. As the remote setting from the landmass, the Mariana Trench provides an opportunity to distinguish marine in situ production from a terrestrial origin of brGDGTs that muddles interpretation of shelf sediments. However, at the current stage, it is unclear how similar and different in brGDGTs-producing microbes as well as their response behaviors to ambient environments between the Mariana Trench and continental shelf sediments. In addition, the weight contribution to brGDGTs from sediments or water column remains elusive. Since factors such as nutrients, particle loading, bacterial community composition, oceanographic parameters (e.g., oxygenation, salinity, currents) vary significantly between the shelf and trench, the organisms to biosynthesize brGDGTs are likely different between two marine settings. Therefore, it should be caution to apply the MBT/CBT and BIT proxies in the open ocean." We also updated our abstract in the revised manuscript.

2) I fundamentally disagree with the use of soil calibrations for reconstructing seawater, or porewater, pH and temperature (lines 360-372). There is no evidence supporting the applicability of these calibrations. Therefore, the brGDGT-based pH reconstructions cannot be used as evidence for in situ production. Specifically, it is currently unclear if the clades of bacteria producing brGDGTs in soils are similar to those in the ocean, particularly because the perceived adaptation of branched GDGT distributions to environmental parameters has previously been suggested to be a community effect.

Response: We accept reviewer's suggestion. It is true that the brGDGT-based pH and MAT reconstructions were established based on terrestrial samples (soil and peat). Thus, MBT, CBT and modified parameters cannot be used as evidence for in situ production. So, we have deleted relevant results and discussion. We also deleted the figure 7 and figure 9b, c, and d that are related to global calibration of brGDGTs-proxies and temperature/pH. Please see lines 15, 137-140, 275-279, 374-380 and 423-427 as well as our response to the reviewer 1.

3) I think that the number of figures in the main text could be reduced in order to streamline the manuscript. I suggest moving figures 3-5 to the supplement.

Response: Since both reviewers mentioned too many figures, and unconvincing calibration of brGDGTs-proxies in ocean, we removed figure 7, 9b, c, and d in the revised manuscript. However, we kept the figs 3-5 because reviewer 1 think the most novelty of our work is the first report on the absence of 5-methyl brGDGTs and strong predominance of 6-methyl brGDGTs in marine sediments. We agree with the reviewer 1's statement. In order to determine the source of brGDGTs, the information on bulk geo-chemical parameters of total organic matter and molecular composition of brGDGTs is greatly needed. Taken together, we still keep the figs. 3-5 in the revised manuscript, but deleted figs. 7 and 9 that are about global calibration of brGDGTs-derived proxies with temperature and pH. The detailed explanation for removal of these figures can be found our response to the reviewer 1's comment.

4) Finally, I recommend the authors carefully examine their manuscript to fix multiple typos and grammar.

Response: Thank you for pointing out this issue. All authors checked the grammars carefully and made the corrections if necessary.

Additional comments: 1) All figures: To ensure accessibility, please use color blind-friendly colors, e.g., do not use red and green in the same figure.

Response: Thank you for the suggestion. We have changed the color of figures.

2) Line 8-24: Please correct grammar issues throughout the abstract

Response: We checked the grammars carefully and made the corrections in abstract. Please see the revised manuscript for the details.

3) Line 16: Please specify that $\delta$13C values are for OC

Response: We have specified $\delta$13C to $\delta$13COC.

4) Line 28-30: Please consider finding more appropriate citations. Sinninghe Damste 2000 and Weijers et al. are neither the first to report on iGDGTs/brGDGTs nor are these the most comprehensive papers.

Response: We have replaced previous citations with Schouten et al., 2013, which is the most comprehensive and cited papers about GDGTs.

5) Line 120-122: Without response factors of GDGTs relative to the C46 standard, concentrations cannot be determined. Please report concentrations as response units or peak areas normalized to OC.

Response: This is a good comment. In the revised manuscript, we added the sentences as "Since all brGDGT isomers were assumed to have an identical response factors on the instrument, our analytical method is better regarded as semi-quantification." (Line 289-290). However, it is common to report the concentration of GDGTs in gram per dry weight sediments or OC even without consideration of response factor in,literatures. So we still keep the current format for the concentration.

---

## Author Response (AR1)

Dear Prof. Silvio Pantoja,

Thank you very much for editorial efforts on our manuscript. My coauthors and I reviewed your comments carefully and found that they are very valuable to improve our manuscript. So, we made changes according to your suggestions. Also, for your convenience, I highlighted all those changes in the revised manuscript. Please see our detailed response as follows.

I am looking forward to hearing your decision or any comments.

Regards-Yunping

1)  **Reviewer 2**.  "The uniqueness of the study site is both a strength and a weakness of the presented work. "

**Editor**: The reviewer is asking to discuss this caveat in the context of your conclusions, not merely state that we "…we have to be cautious…". Please elaborate further on the new version.

**Response**: This is a good comment. In the revised manuscript, we added the content in the discussion and conclusion. From line 344-356 (sec. 4.4), we added a paragraph as "**The unique composition of brGDGTs in the Mariana Trench has significant implications on the brGDGTs-based proxies. As a remote setting from the landmass, the Mariana Trench provides an opportunity to distinguish marine in situ production from a terrestrial origin of brGDGTs that usually muddles the interpretation of shelf sediments. However, it is unclear what the similarity and difference are for brGDGTs-producing microbes and their response to environmental factors between the Mariana Trench and continental shelf. In addition, the weight contribution to the brGDGT pool from sediments and water column remains elusive. Since factors such as nutrients, particle loading, bacterial community, and oceanographic parameters (e.g., oxygenation, salinity, currents) vary significantly between the shelf and trench as well as among different hadal trenches, the brGDGT-producing microbes are likely different. Therefore, the investigation of brGDGTs in multiple hadal trenches and shallow marine regions are needed to decipher their source and environmental control, that are beneficial for accurate application of the brGDGTs-based proxies such as MBT, CBT and BIT**."

We also added two sentences in the conclusion as "**4)  The uniqueness of the Mariana Trench that is remote from any landmass allows distinguishing marine in situ production from a terrestrial origin of brGDGTs. However, it is unclear how comparable this unique site is to shallow marine settings and other hadal trenches. Therefore, the comparison studies of brGDGTs for different hadal trenches as well as between hadal and non-hadal sites are recommended**." (line 377-381)

In the abstract, we added a sentence "**Given the uniqueness of the Marina Trench, we recommend more studies for different trenches and shallow regions in order to holistically understand the biosynthesis and environmental control of brGDGTs**." (line 24-26)

2)  **Reviewer 2**. Line 28-30: Please consider finding more appropriate citations. Sinninghe Damste 2000 and Weijers et al. are neither the first to report on iGDGTs/brGDGTs, nor are these the most comprehensive papers.

Authors: We have replaced previous citations with Schouten et al., 2013, which are the most comprehensive and cited papers about GDGTs.

**Editor**: Please revise. A 2013 paper is not older than a 2000 paper. Which one is the first to report on iGDGTs/brGDGTs?

**Response**: We searched the references and added two suitable references here. De Rosa and Gambacorta (1988) is a classic paper about lipids of archaebacteria and introduced isoprenoid tetraethers (this paper has been cited by more than 700 times). Sinninghe Damsté et al. (2000) is the first report for non-isoprenoid dialkyl diglycerol tetraether lipids (BrGDGT) in sediments. The detailed information about these two references are:

1) De Rosa, M., Gambacorta, A., 1988. The lipids of archaebacteria. Prog. Lipid Res. 27, 153-175. https://doi.org/10.1016/0163-7827(88)90011-2.
2) Sinninghe Damsté, J.S., Hopmans, E.C., Pancost, R.D., Schouten, S., Geenevasen, J.A.J., 2000. Newly discovered non-isoprenoid dialkyl diglycerol tetraether lipids in sediments. Chem. Commun. 23, 1683–1684, https://doi.org/10.1039/B004517I.

3) **Reviewer 2, "5)** Line 120-122: Without response factors of GDGTs relative to the C46 standard, concentrations cannot be determined. Please report concentrations as response units or peak areas normalized to OC.
Authors: "This is a good comment. In the revised manuscript, we added the sentences as Since all brGDGT isomers were assumed to have an identical response factors on the instrument, our analytical method is better regarded as semi-quantification." (Line 289-290). "However, it is common to report the concentration of GDGTs in … without consideration of response factor in literatures. So, we still keep the current format for the concentration. "
**Editor:** "it is common to report the concentration of GDGTs … without consideration of response factor"). This is not correct. Is there much variability in response factors for brGDGT isomers?

**Response**: This is a good comment. As the reviewer and the editor pointed out, we did not consider the response factors. In our study, the absolute concentration is actually not important, since we only discussed the composition of brGDGTs and derived proxies (such as BIT, CBT and MBT), all of which are based on relative abundance of brGDGTs. The reason we presented the absolute concentration is for the followers who may want to compare their GDGT concentrations with ours. In the revised manuscript, we accepted the reviewer's and editor's suggestion and removed the contents related to absolute concentrations. Since our original quantification is by an internal standard (C46 GDGT) method, and thus we did not measure the exact total volume of each sample loaded on HPLC which is not needed for the internal standard method. Given these facts, we rewrote the first paragraph of Sec. 3.2. We only reported the relative abundance of GDGTs based on peak area of each compound.

In sum, we changed the contents of Sec. "2.2 Lipid extraction and GDGT analyses" as well as "3.2 Concentration and composition of GDGTs". From line 111 to 114, we wrote the sentence as "**Since the response factors of GDGTs were not determined due to the lack of authentic standard, we did not calculate the absolute concentration of GDGTs. Instead, we reported the relative concentration based on peak areas (pa) of respective GDGTs normalized to total GDGTs.**" From line 173-179, we wrote as "**The fractional abundance of iGDGTs and brGDGTs were summarized in Table 2. iGDGTs were the dominant components, accounting for 96.8% to 98.6% of total GDGTs in Mariana Trench sediments. The proportion of brGDGTs was substantially lower than that of iGDGTs, ranging from 1.4%**

**to 3.2%. For all sediment samples, the BIT index remained at a low level (0.03 ± 0.01).".**

4) Reviewer 1. "3. The application of soil pH index CBT' and mean annual air temperature index MATmr in this marine setting is unconvincing. "
Authors: " As mentioned above, we accepted the reviewer's suggestion and removed all relevant figures and discussion."
Editor: Being that the case, the associated conclusion should be removed from the abstract as well (line 15).

**Response**: we accepted this suggestion and already deleted "This phenomenon may reflect an adaption of brGDGTs-producing bacteria to weak alkaline and low temperature conditions" in the abstract.

Minor comment:
1. Line 30. Change IGDTs with iGDGTs

**Response**: we changed "IGDGTs" into "iGDGTs" in the revised manuscript.

2. Line 15. Replace "Rota Evaporator" with "rotary evaporator "

**Response**: we changed "Roto Evaporator" into "rotary evaporator" in the revised manuscript.

3. Line 175.? Why "stated elsewhere"?

**Response**: In order to avoid confusing, we deleted "stated elsewhere". In the revised manuscript, we wrote as "The significance level was set at $p < 0.05$."

4. Line 190. Consider using gdw instead of "dry weight sediment (dws) " since it is more commonly used

**Response**: This is a good point and we accept it.

5. Figure 4. Remove comma in Luo et al., (2017)

**Response**: we already deleted comma here in the revised manuscript.

---

## Editor Decision (ED1)

February 24, 2020

Review of Predominance of hexamethylated 6-methyl branched glycerol dialkyl glycerol tetraethers in the Mariana Trench: Source and environmental implication" *by* Wenjie Xiao et al. (bg-2019-391)

Dear Dr. Xu,

Thanks for submitting responses to reviewers' comments to your article. Both reviewers are very positive about this article, and I agree with them as well.

I am looking forward to reviewing a new version that incorporates those comments. Please pay attention, particularly to the following aspects:

1) Reviewer 2.  "The uniqueness of the study site is both a strength and a weakness of the presented work. " The reviewer is asking to discuss this caveat in the context of your conclusions, not merely state that we "…we have to be cautious…". Please elaborate further on the new version.

2) Reviewer 2. Line 28-30: Please consider finding more appropriate citations. Sinninghe Damste 2000 and Weijers et al. are neither the first to report on iGDGTs/brGDGTs, nor are these the most comprehensive papers.

   Authors: We have replaced previous citations with Schouten et al., 2013, which are the most comprehensive and cited papers about GDGTs.

   Editor: Please revise. A 2013 paper is not older than a 2000 paper.  Which one is the first to report on iGDGTs/brGDGTs?

3) Reviewer 2, "5) Line 120-122: Without response factors of GDGTs relative to the C46 standard, concentrations cannot be determined. Please report concentrations as response units or peak areas normalized to OC.

   Authors: "This is a good comment. In the revised manuscript, we added the sentences as *Since all brGDGT isomers were assumed to have an identical response factors on the instrument, our analytical method is better regarded as semi-quantification.*" (Line 289-290). "However, it is common to report the concentration of GDGTs in … without consideration of response factor in literatures. So we still keep the current format for the concentration. "

   Editor:  "it is common to report the concentration of GDGTs … without consideration of response factor").  This is not correct. Is there much variability in response factors for brGDGT isomers?

4) Reviewer 1. "3. The application of soil pH index CBT' and mean annual air temperature index MATmr in this marine setting is unconvincing. "

Authors: " As mentioned above, we accepted the reviewer's suggestion and removed all relevant figures and discussion."

Editor: Being that the case, the associated conclusion should be removed from the abstract as well (line 15).

Other comments from Associate Editor:

1. Line 30. Change IGDTs with iGDGTs

2. Line 15. Replace "Rota Evaporator" with "rotary evaporator "

3. Line 175.? Why "stated elsewhere"?

4. Line 190. Consider using gdw instead of "dry weight sediment (dws) " since it is more commonly used

5. Figure 4. Remove comma in Luo et al., (2017)

Looking forward to hearing from you

Best regards

Silvio Pantoja-Gutiérrez

Associate editor

---

## Author Response (AR2)

Dear Prof. Silvio Pantoja,

Thanks for your comment. After two round reviews, the article is accepted except for a minor comment about the abstract: it is better to remove the sentence "Given the uniqueness of the Marina Trench, we recommend more studies for different trenches and shallow regions in order to holistically understand the biosynthesis and environmental control of brGDGTs."(line 24-26) since it is such a general statement that does not belong to the Abstract section. This is a good suggestion, So I removed the sentence in line 24-26: Please see the revised manuscript for the details.

Regards-Yunping

Yunping Xu, Dr.
Professor of College of Marine Sciences
Shanghai Ocean University, Shanghai, China
email: ypxu@shou.edu.cn
Phone: 86-15652917880
http://www.researcherid.com/rid/F-8042-2011